# Voltage Stability Analysis of a Power System with Wind Power Based on the Thevenin Equivalent Analytical Method

**Xia Zhou** [1,*], **Yishi Liu** [1], **Ping Chang** [2], **Feng Xue** [3] **and Tengfei Zhang** [1]

1    College of Automation and Artificial Intelligence, Nanjing University of Posts and Telecommunications, Nanjing 210023, China; 1220055817@njupt.edu.cn (Y.L.); tfzhang@126.com (T.Z.)
2    School of Electrical Engineering, Southeast University, Nanjing 210018, China; changping@seu.edu.cn
3    NARI Group Corporation, State Grid Electric Power Research Institute, Nanjing 211106, China; xue-feng@sgepri.sgcc.com.cn
*    Correspondence: zhouxia@njupt.edu.cn; Tel.: +86-138-1390-0451

**Abstract:** The traditional black-box Thevenin equivalent method cannot analyze the influence mechanism of wind power integration on Thevenin equivalent parameters. With the increase in wind power penetration, it will be difficult to accurately assess the voltage stability of power systems with wind power. Therefore, a Thevenin equivalent analytical method is proposed to analyze the voltage stability of a power system with wind power. This method adopts the equivalent model of wind power integration based on the current source model. It establishes the Thevenin equivalent analytical model for power systems with wind power by dividing the node types. Then, the analytical expressions of the Thevenin equivalent parameters are derived based on two equivalent modes to characterize the mechanism of wind power integration on the Thevenin equivalent parameters. In addition, the calculation flow chart of the voltage stability criterion is formulated based on the analytical value of the Thevenin equivalent impedance under different load growth ratios and wind power penetrations. Finally, a case study is conducted on the improved IEEE 39 node system with wind power. The results demonstrate the feasibility and effectiveness of the proposed Thevenin equivalent analytical method, which can more accurately judge the voltage stability of the power system with wind power.

**Keywords:** wind power system; voltage stability; Thevenin equivalent analytical method; load multiplier; wind power penetration





## 1. Introduction

Because of the large scale and the complicated operation mode of power systems, comprehensive and refined simulation modeling is not suitable for the online rapid and accurate evaluation of system voltage stability [1]. With the development and application of wide-area measurement technologies such as phasor measurement units (PMU), multi-source power data are effectively collected. In view of the influence of new energy grid connections on the voltage stability of power systems, the analysis of the voltage stability of new energy power systems has become one of the current research hotspots [2–4]. Reference [2] compares and analyzes the voltage stability characteristics of wind farms under different control strategies and provides a theoretical summary guidance. Reference [3] analyzed the active power dynamic characteristics of a grid under voltage control under large-scale wind power penetration and systematically introduced the influence of grid stability characteristics on the grid. Reference [4] proposed a control strategy to improve voltage stability based on TCSC–STATCOM. A large number of studies have shown that voltage stability characteristics directly reflect the operation safety of large power grids. In order to quickly and accurately evaluate the voltage stability of new power systems, the Thevenin equivalent method is widely used. This method essentially simplifies the nonlinear power system to the Thevenin equivalent circuit of the critical node. It then

analyzes the modulus relationship between the Thevenin equivalent impedance of the equivalent node and the load impedance according to the principle of maximum power transmission to judge the voltage stability of the system [5–7]. Reference [5] used the Thevenin equivalent method to calculate the parameters of the VRB equivalent model to establish the ESS control model. References [6,7] all used Thevenin's equivalent method to study, observe the system active power change and realize the voltage stability control. Among them, when the voltage stability is studied based on the Thevenin equivalent method, the change of the impedance mode is closely related to the load characteristics of the equivalent node and the operation mode of the system power supply. Therefore, it is particularly important to study the classification of grid nodes and analyze the impact of new energy units on equivalent nodes. However, the classification of grid nodes and the qualitative analysis of new energy nodes in the above literature are weak.

The output characteristics of intermittent new energy sources such as wind power generations and photovoltaic generations are very different from traditional units. The increase in their grid-connected capacities will continuously change the operation mode of the system power supply. It will have a particular nonlinear and uncertain influence on the calculation of the Thevenin equivalent impedance and other parameters [8,9]. Therefore, when studying the voltage stability of power systems with wind power based on Thevenin equivalent theory, it is necessary to further establish the Thevenin equivalent parameter identification and evaluation method that takes into account the characteristics of wind power and to study the mechanism of wind power connection on Thevenin equivalent parameters, thereby increasing the accuracy of system voltage stability assessments.

At present, according to the different solving principles, the identification methods of the Thevenin equivalent parameters of the power systems with wind power can be divided into the following two categories:

(1) The first identification method is based on the results of power flow calculations. Firstly, the PMU is used to obtain the node voltage and the current data of single or multiple time sections after the system power flow calculation. Then, the two-point method, total differential, deviation correction and other methods are used to process the data. However, the processing methods have their limitations. For example, the two-point method has problems such as parameter drift and insensitivity to parameter disturbances in the equivalent system. The identification accuracy of the total differential is affected by the initial value of the calculation. The online application of the deviation correction is limited by the PMU sampling time [10–12]. In addition, for power systems with wind power, the first type of identification method has a common blemish. It directly includes intermittent wind power and nonlinear loads in the equivalent network in the form of a black box, which will lead to a decrease in the accuracy of the identification and evaluation of the Thevenin equivalent parameters. Additionally, it is also unable to characterize the influence mechanism and action mechanism of wind power, load and other factors on the Thevenin equivalent parameters [13]. Furthermore, this will affect the accuracy of the evaluation of the voltage stability of the equivalent node and will cause the incapability to give subsequent control strategies to the problem of the voltage stability of the equivalent node based on the pattern of variations of the Thevenin equivalent parameters affected by wind and load.

(2) The second identification method is based on the analysis of the system network. Firstly, based on the node voltage equation, by dividing the network node types, the analytical equivalent circuit of the Thevenin equivalent model is derived. The mechanism of the generator nodes and load nodes on the Thevenin equivalent parameter is analyzed. Then, the two approximate analytical equivalent methods of the Thevenin equivalent parameters are studied [13–15]. Considering the coupling effect of the equivalent node branch, the coupling effect term is equivalent to the Thevenin equivalent model impedance or potential. This kind of method can probe and characterize the influence mechanism of the generator nodes and load nodes

on the voltage stability of an equivalent node. However, in the power system with wind power, to analyze the influence mechanism of the wind power connection on the voltage stability of the equivalent nodes, it is necessary to further consider the influence of the wind power nodes on the Thevenin equivalent parameters and to characterize their role. The voltage stability of the power system containing wind power can be judged more accurately.

Based on the above statements, starting from the basic Thevenin equivalent model, this paper fully considers the influence of new energy units in the new power system on the parameters of the equivalent nodes, and analyzes the shortcomings of the existing Thevenin equivalent models for power systems with wind power. Then, based on the system network analytical identification method, a Thevenin equivalent analytical method for a power system with wind power is proposed. The equivalent value of the wind turbine is the injection current source, the current increment is introduced into the equivalent node model after the grid connection and the Thevenin equivalent model of the power system including wind power is established by using the admittance matrix. Finally, a simulation study is conducted on the IEEE 39-bus system with wind power, which verifies the feasibility and effectiveness of the proposed method.

## 2. Basic Principles of Thevenin Equivalence in Power System

### 2.1. The Basic Model of Thevenin Equivalent

Reference [13] pointed out that any complex power system can be represented using a load bus and a Thevenin equivalent seen from the load bus. Although the equivalent part may consist of a number of generators, transformers, transmission lines and loads, it can be described by only two parameters of $E_{th}$ and $Z_{th}$, namely, the Thevenin equivalent voltage and impedance. The Thevenin equivalent process of the power system is shown in Figure 1. At any time, looking into the system from a certain load node $k$, the rest of the power system can be equivalent to a voltage source series impedance to supply power to the load node $k$ [16]. The Thevenin equivalent system is a simple single-input single-output two-node system, which effectively simplifies the complex power system.

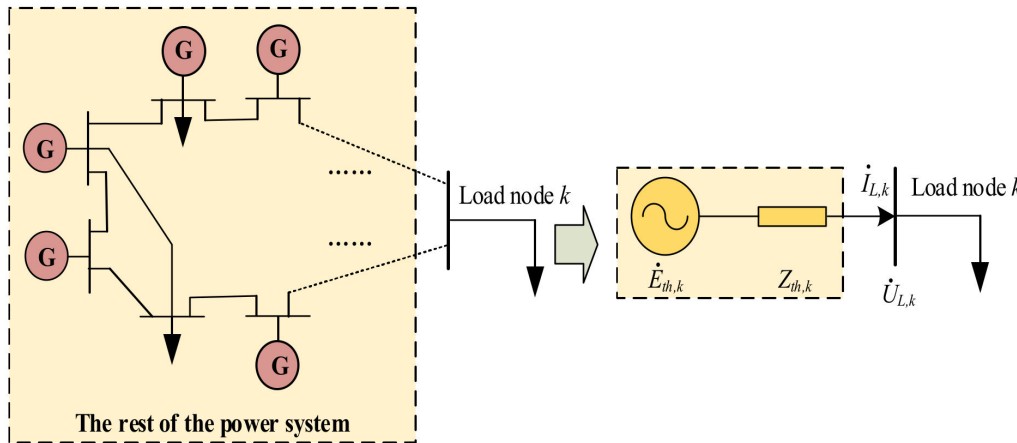

**Figure 1.** The schematic diagram of Thevenin equivalent for the power system.

In Figure 1, $\dot{E}_{th,k}$ and $Z_{th,k}$ are the Thevenin equivalent potential and Thevenin equivalent impedance of load node $k$. $\dot{U}_{L,k}$ and $\dot{I}_{L,k}$ are the voltage phasor and current phasor of load node $k$, respectively. Then, the basic model of the Thevenin equivalent is:

$$\dot{U}_{L,k} = \dot{E}_{th,k} - Z_{th,k} \times \dot{I}_{L,k} \tag{1}$$

The Thevenin equivalent parameters $\dot{E}_{th,k}$ and $Z_{th,k}$ will vary along with the change of the power supply operation mode and network structure of the rest parts of the power system.

*2.2. Voltage Stability Criterion Based on Impedance Modulus Ratio*

The calculation scheme of the voltage stability criterion based on the impedance modulus ratio is simple, and the physical concept is evident [5]. In this paper, this criterion is utilized to analyze and evaluate the voltage stability of the medium load node *k* in Figure 1. The calculation method is:

$$L_k = 1 - \frac{\left|Z_{th,k}\right|}{\left|Z_{L,k}\right|} \tag{2}$$

where $Z_{L,k}$ is the load impedance of the equivalent node *k*. When $0 < L_k < 1$, the equivalent node voltage is stable. The closer $L_k$ is to 0, the worse the voltage stability of the equivalent node. When $L_k < 0$, the voltage of the equivalent node is unstable.

The load impedance can be calculated from the voltage and current phasors of the equivalent nodes, namely:

$$Z_{L,k} = \frac{\dot{U}_{L,k}}{\dot{I}_{L,k}} \tag{3}$$

The key to the criterion of voltage stability based on the impedance modulus ratio is the identification and calculation of the Thevenin equivalent impedance parameters.

*2.3. Insufficiency of the Existing Thevenin Equivalent Model of Wind Power System*

With a high proportion of renewable energy sources such as wind power and photovoltaics being connected to the power system, the power supply operation mode and network structure will also undergo tremendous changes. Taking new energy wind power as an example, when the penetration rate of the wind power is high, it will be directly classified into the rest parts of the power system if the wind power is treated as a black box, as shown in Figure 1. Then, the first type of identification method is used to calculate the Thevenin equivalent parameters of the equivalent node. On the one hand, due to the intermittent and fluctuating nature of wind power generation, the time-varying nonlinearity of the power system in practical applications makes it difficult to identify the Thevenin equivalent parameters accurately [13]. On the other hand, the mechanism of the wind power connection on Thevenin's equivalent parameters cannot be analyzed and characterized, and it is difficult to analyze the mechanism of influence on Thevenin's equivalent parameters when the wind power is connected in different proportions. This will affect the accurate assessment of the voltage stability of the power system with wind power.

Reference [17] proposes that for power systems with wind power, the wind power in the system can be equivalent to a voltage source in the form of series impedance, and in parallel at the equivalent load node *k* at the same time, as shown in Figure 2. In the equivalent system, the Thevenin equivalent parameters $\dot{E}_{Wth}$ and $Z_{Wth}$ of wind power are equivalent according to the Thevenin equivalent parameters $\dot{E}_{Gth}$ and $Z_{Gth}$ of the synchronous generators of traditional power plants. This model can quantitatively analyze the relationship between the proportion of wind power integration and the static voltage stability margin of the system from the perspective of the enormous power grid. However, conventional synchronous generators are used to replace wind turbines, ignoring the difference between the operating characteristics of wind turbines and traditional synchronous generators. The accuracy of the obtained Thevenin equivalent parameters is low, which will reduce the accuracy of the voltage stability assessment of the power system with wind power.

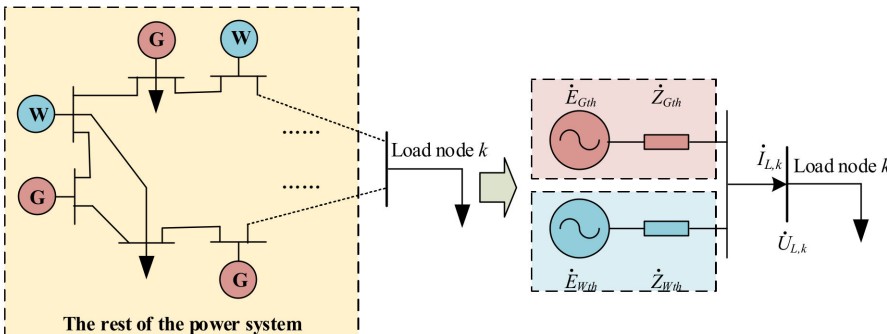

**Figure 2.** The Thevenin equivalent model of wind power integration based on the traditional synchronous unit model.

## 3. Thevenin Equivalent Analysis Method for the Power System with Wind Power

Aiming at the shortcomings of the existing Thevenin equivalent models for power systems with wind power, this chapter first analyzes the applicable wind power grid-connected models, takes the wind power grid-connected equivalent as the injection current source, and analyzes the wind power injection current mechanism. Then, according to the node types of the wind power grid-connected system, a Thevenin equivalent analytical model of the power systems with wind power is established. The analytical expressions of the Thevenin equivalent parameters are derived according to two equivalent methods, and the mechanism and influencing factors of the wind power connection on the Thevenin equivalent parameters are theoretically studied. Finally, the corresponding voltage stability criterion is calculated using the analytical value of the Thevenin equivalent impedance, and the voltage stability of the system under different load growth multiples and different wind power penetration rates is evaluated. Figure 3 is a schematic diagram of the Thevenin equivalent analysis method for wind power systems in this chapter.

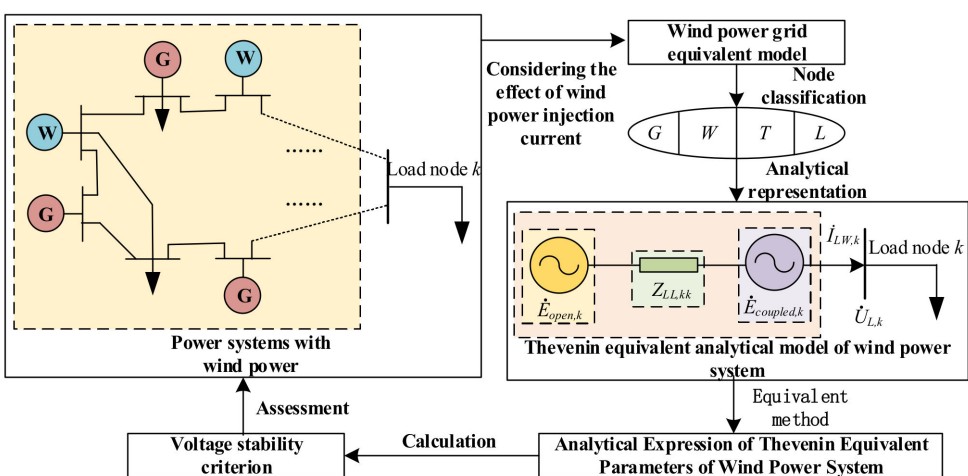

**Figure 3.** The schematic diagram of Thevenin equivalent analysis method for power system with wind power.

### 3.1. Equivalent Model of Wind Power Grid Connection

The voltage stability of a power system mainly depends on the voltage stability of key load nodes in the system [16]. To study the voltage stability of a power system with wind power, it is necessary to further consider the influence of wind power integration on the voltage stability of the load nodes.

Considering the update and improvement of the wind turbine control mode and the equipment configuration of the automatic voltage control system, wind power will have specific active and voltage control capabilities [18,19]. The study of the system load node

voltage after wind power is connected to the grid needs to calculate the current flowing through the load node itself, so as to use Ohm's law to obtain the coupling effect of the wind power node to the load node. A centralized wind power node can be equivalent to an injected current source. Then, the injected current of the wind power node can be converted into the injected current increment of the load node through the impedance matrix, so as to study the voltage stability of the load node after the wind power is connected, as shown in Figure 4.

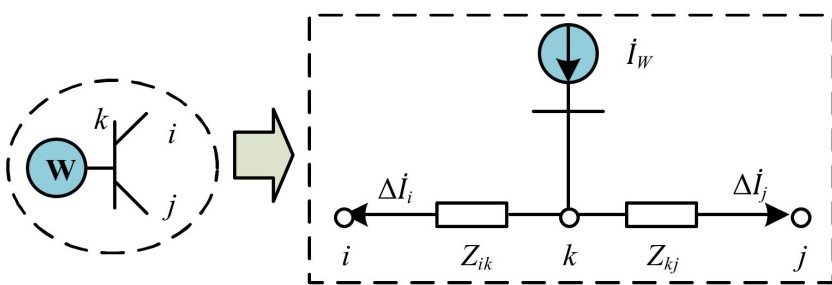

**Figure 4.** The equivalent model of wind power integration based on the injection current source model.

In Figure 4, $Z_{ik}$ and $Z_{kj}$ are the impedances between nodes $i$, $k$ and nodes $k$, $j$, respectively. When wind power is connected to the grid in the form of the current source $\dot{I}_W$ at node $k$, the increments of the injected current $\Delta \dot{I}_i$ and $\Delta \dot{I}_j$ are generated at node $i$ and node $j$, respectively. Among them:

$$\Delta \dot{I}_i = \frac{Z_{ik}}{Z_{ik} + Z_{kj}} \dot{I}_W \tag{4}$$

$$\Delta \dot{I}_j = \frac{Z_{kj}}{Z_{ik} + Z_{kj}} \dot{I}_W \tag{5}$$

### 3.2. An Analytical Model of Thevenin Equivalence for the Power System with Wind Power

According to the current increment obtained by the wind power model, an analytical model is established using Thevenin's Equivalence Theorem. The node types of wind power systems can be divided into four categories: generator node, wind power node, connection node and load node. Assuming that the direction of the current flowing into the generator node is the positive direction, and considering that the injected current of the connection node is zero, the node voltage equation $I = YU$ is expanded according to the node type to obtain:

$$\begin{bmatrix} I_G \\ I_W \\ 0 \\ -I_L \end{bmatrix} = \begin{bmatrix} Y_{GG} & Y_{GW} & Y_{GT} & Y_{GL} \\ Y_{WG} & Y_{WW} & Y_{WT} & Y_{WL} \\ Y_{TG} & Y_{TW} & Y_{TT} & Y_{TL} \\ Y_{LG} & Y_{LW} & Y_{LT} & Y_{LL} \end{bmatrix} \begin{bmatrix} U_G \\ U_W \\ U_T \\ U_L \end{bmatrix} \tag{6}$$

Among them, $I$ and $U$ represent the current and voltage vectors of all nodes, respectively, $Y$ represents the node admittance matrix, and the subscripts $G$, $W$, $T$ and $L$ are used to represent the generator nodes, wind power nodes, connection nodes and load nodes, respectively, so $I$, $U$ and $Y$ are divided into blocks.

The system of the equations in Equation (6) is expanded and eliminated according to the equation form of Equation (1), and the voltage vectors $U_W$ and $U_T$ of the wind power node and the connecting node are eliminated to obtain:

$$U_L = E_{open} - Z_{LL}I_{LW} \tag{7}$$

Among them:

$$E_{open} = Z_{LL}\left(Y_c Y_f^{-1} Y_e - Y_b\right)U_G \tag{8}$$

$$Z_{LL} = \left(Y_a - Y_c Y_f^{-1} Y_d\right)^{-1} \tag{9}$$

$$I_{LW} = I_L + \Delta I_L \tag{10}$$

$$\Delta I_L = Y_c Y_f^{-1} I_W \tag{11}$$

$$\begin{aligned}
Y_a &= Y_{LL} - Y_{LW}Y_{TW}^{-1}Y_{TL}, \; Y_b = Y_{LG} - Y_{LW}Y_{TW}^{-1}Y_{TG} \\
Y_c &= Y_{LT} - Y_{LW}Y_{TW}^{-1}Y_{TT}, \; Y_d = Y_{WL} - Y_{WW}Y_{TW}^{-1}Y_{TL} \\
Y_e &= Y_{WG} - Y_{WW}Y_{TW}^{-1}Y_{TG}, \; Y_f = Y_{WT} - Y_{WW}Y_{TW}^{-1}Y_{TT}
\end{aligned} \tag{12}$$

In the formula, $E_{open}$ represents the open-circuit voltage vector oriented to the load nodes, $Z_{LL}$ is the impedance matrix trained to load nodes, $\Delta I_L$ is the injected current increment converted from the wind turbine node injected current $I_W$ to the load node through the matrix $Y_c Y_f^{-1}$, $I_{LW}$ is the node current vector after the load node is calculated with $\Delta I_L$, and $Y_a \sim Y_f$ has no special meaning and is only a simplified expression.

Substituting the load node $k$ as the equivalent node into Equation (7), the Thevenin equivalent analytical model of the power system with wind power can be obtained as:

$$\dot{U}_{L,k} = \dot{E}_{open,k} - Z_{LL,kk}\dot{I}_{LW,k} - \dot{E}_{coupled,k} \tag{13}$$

Among them:

$$\dot{E}_{coupled,k} = \sum_{l=1,l\neq k}^{n} Z_{LL,kl}\dot{I}_{LW,l} \tag{14}$$

Figure 5 is the circuit diagram of the Thevenin equivalent analytical model of the power system with wind power, which mainly includes three items:

(1) The open-circuit voltage $\dot{E}_{open,k}$ of the load node $k$ is the voltage phasor of the load node $k$ when all the branches where all the load nodes are open. It can be seen from Equation (8) that this item reflects the influence of the voltage of each generator node on the equivalent potential, which can be partially corresponding to $\dot{E}_{th,k}$ in the Thevenin equivalent basic model.

(2) The self-impedance $Z_{LL,kk}$ of the load node $k$ can be seen by Equation (9), which considers the network topology information of the wind power node access and can be partially corresponding to $Z_{th,k}$ in the Thevenin equivalent basic model.

(3) The coupling effect voltage drop $\dot{E}_{coupled,k}$ of the load node $k$, which takes into account the cumulative coupling effect caused by voltage drop on the corresponding mutual impedance caused by the injected current of the other load nodes and the injected current increment of the wind power nodes. According to the research, it is shown that part of the equivalent potential should be equal to the Thevenin equivalent potential $\dot{E}_{th,k}$ and the other amount should be equal to the Thevenin equivalent impedance $Z_{th,k}$, but the proportion of each part of the equivalent is not precise.

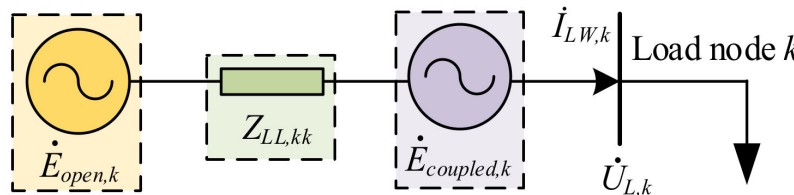

**Figure 5.** The circuit diagram of Thevenin equivalent analytical model for the power system with wind power.

### 3.3. Analytical Expression of Thevenin Equivalent Parameters of Power System with Wind Power

According to Reference [13], in this paper, the coupling effect term $\dot{E}_{coupled,k}$ is addressed as follows: (1) all of them are equivalent to the electric potential, and (2) all of them are equivalent to the impedance. The analytical expression of the Thevenin equivalent parameters of the power system with wind power were derived in two equivalent ways.

(1)  Equivalent mode 1

The coupling effect term $\dot{E}_{coupled,k}$ is equivalent to the electric potential, and the Thevenin equivalent circuit diagram of the power systems with wind power in equivalent mode one is shown in Figure 6.

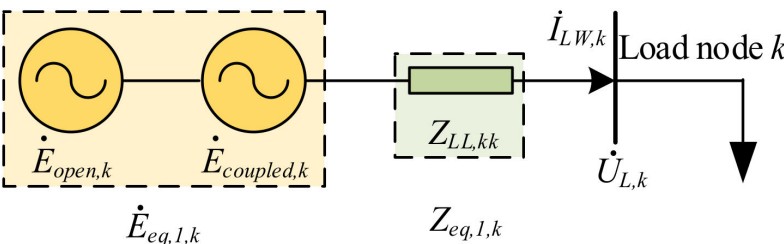

**Figure 6.** The Thevenin equivalent circuit diagram of the power system with wind power in equivalent mode 1.

Then, the Thevenin equivalent analytical model of the power system with wind power under equivalent mode 1 is:

$$\dot{U}_{L,k} = \dot{E}_{eq,1,k} - Z_{eq,1,k}\dot{I}_{LW,k} \tag{15}$$

Among them:

$$\dot{E}_{eq,1,k} = \dot{E}_{open,k} - \dot{E}_{coupled,k} \tag{16}$$

$$Z_{eq,1,k} = Z_{LL,kk} \tag{17}$$

In Equation (15), $\dot{E}_{eq,l,k}$ and $Z_{eq,l,k}$ represent the analytical values of the Thevenin equivalent potential and Thevenin equivalent impedance of the load node $k$ under equivalent mode 1. Among them, $\dot{E}_{eq,l,k}$ will be affected by the coupling effect and $Z_{eq,l,k}$ in the equation is only related to the physical structure and parameters of the power system network topology. If the structure and associated parameters of the system network topology remain unchanged after the wind power integration, $Z_{eq,l,k}$ will remain unchanged.

(2)  Equivalent mode 2

At this time, the coupling effect term $\dot{E}_{coupled,k}$ is equivalent to impedance, and the Thevenin equivalent circuit diagram of the power system with wind power in equivalent mode two is shown in Figure 7.

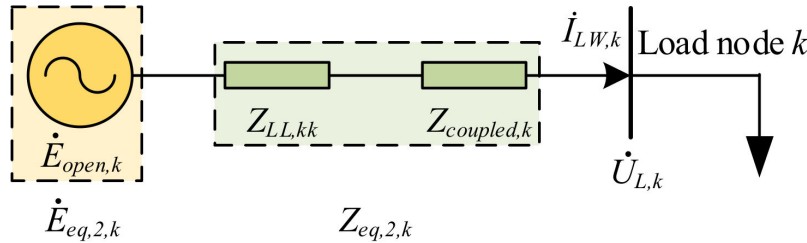

**Figure 7.** The Thevenin equivalent circuit diagram of the power system with wind power in equivalent mode 2.

Then, the Thevenin equivalent analytical model of the power system with wind power under equivalent mode two is:

$$\dot{U}_{L,k} = \dot{E}_{eq,2,k} - Z_{eq,2,k}\dot{I}_{LW,k} \tag{18}$$

Among them:

$$\dot{E}_{eq,2,k} = \dot{E}_{open,k} \tag{19}$$

$$Z_{eq,2,k} = Z_{LL,kk} + Z_{coupled,k} \tag{20}$$

$$Z_{coupled,k} = \frac{\dot{E}_{coupled,k}}{\dot{I}_{LW,k}} = \sum_{l=1,l\neq k}^{n} Z_{LL,kl}\frac{\dot{I}_{LW,l}}{\dot{I}_{LW,k}} \tag{21}$$

In Equation (18), $\dot{E}_{eq,2,k}$ and $Z_{eq,2,k}$ represent the analytical values of the Thevenin equivalent potential and Thevenin equivalent impedance of the load node $k$ under equivalent mode 2. The $\dot{E}_{eq,2,k}$ is not only related to the structure of the system network topology and related parameters, but it is also related to the voltage of the generator node in the system. $Z_{eq,2,k}$ is affected by the coupling effect impedance $Z_{coupled,k}$, and $Z_{coupled,k}$ is affected by the injected current of the other load nodes considering the effect of wind power.

*3.4. Voltage Stability Criterion Based on the Analytical Value of the Thevenin Equivalent Impedance*

According to Section 3.3, two analytical expressions $Z_{eq,l,k}$ and $Z_{eq,2,k}$ of the Thevenin equivalent impedance parameters of the wind power system can be obtained, namely, Equations (17) and (20). Substituting $Z_{eq,l,k}$ and $Z_{eq,2,k}$ representing the Thevenin equivalent impedance $Z_{th,k}$ into Equation (2), two voltage stability criteria based on the analytical value of the Thevenin equivalent impedance can be obtained. Among them, the voltage stability criterion $L_{eq,l,k}$ of the equivalent node $k$ under equivalent mode one is:

$$L_{eq,1,k} = 1 - \frac{\left|Z_{eq,1,k}\right|}{\left|Z_{L,k}\right|} \tag{22}$$

$Z_{eq,l,k}$ mainly changes along with the change of the load impedance modulus of equivalent nodes.

The voltage stability criterion $L_{eq,2,k}$ of the equivalent node $k$ under equivalent mode 2 is:

$$L_{eq,2,k} = 1 - \frac{\left|Z_{eq,2,k}\right|}{\left|Z_{L,k}\right|} \tag{23}$$

In order to prove the feasibility and effectiveness of the analytical method in this section, this paper sets different load growth multiples and wind power penetrations to change the power supply operation mode and network architecture of the power system, and calculates the analytical values of the Thevenin equivalent impedance under the two equivalent modes, analyzes the dynamic change law and calculates the corresponding voltage stability criterion. Figure 8 shows the calculation flow charts of the two voltage stability criteria based on the analytical value of the Thevenin equivalent impedance.

Step 1: set the load growth multiple and wind power penetration of the power system with wind power;

Step 2: the node admittance matrix of the system is calculated and divided into blocks according to the node type. Meanwhile, after the power flow calculation, the voltage and current phasors of each node of the system are measured by the PMU;

Step 3: use the relevant data obtained in Step 2 to calculate the parameters $\boldsymbol{E}_{open}$, $\boldsymbol{Z}_{LL}$ and $\boldsymbol{I}_{LW}$ of the Thevenin equivalent analytical model of the power system with wind power and the load impedance $Z_{L,k}$ of the equivalent node $k$;

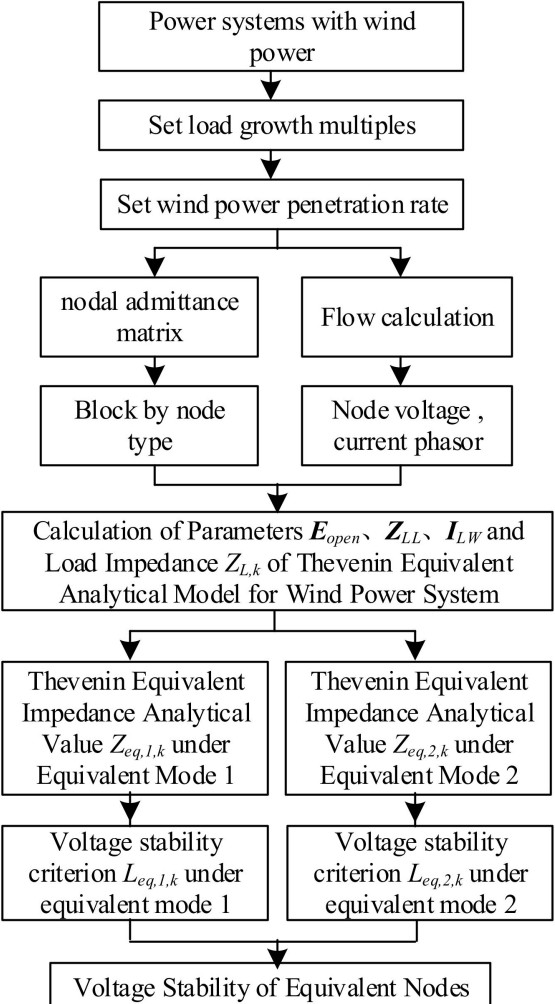

**Figure 8.** The flow chart of voltage stability criterion based on the analytical value of Thevenin equivalent impedance.

Step 4: according to Equations (17) and (20), calculate the analytical values of the Thevenin equivalent impedance $Z_{eq,1,k}$ and $Z_{eq,2,k}$ of the equivalent node $k$ under two equivalent modes;

Step 5: according to Equations (22) and (23), two voltage stability criteria $L_{eq,1,k}$ and $L_{eq,2,k}$ based on the Thevenin equivalent impedance analytic value are calculated to evaluate the voltage stability of the equivalent node at this time.

In Step 1, the load growth multiple changes as the active power and reactive power of all the load nodes in the system increase proportionally, and the power factor remains unchanged. At the same time, the active power output of the generator with equal power is increased. The active power increment of each generator node (except the balance node) is distributed according to the active power ratio of its initial access, and the balance node bears the deviation of the network loss [20].

In this paper, wind power penetration is defined as the ratio of the total installed wind power capacity in the power system to the sum of the active power of the load [21]. In Step 1, the change mode of the wind power penetration is to increase the active power of all the wind power nodes in the system proportionally according to the initial active power ratio of the connected wind power, and replace the active power output of the generator with equal power. Each replaced node (except the balanced node) is associated according to the active power ratio of its initial access, and the error of the network is offset by the balance node.

## 4. Case Study

### 4.1. Simulation Settings

This paper simulates and verifies the feasibility and effectiveness of the proposed analytical method in an improved IEEE 39-bus system with wind power. The system structure is shown in Figure 9. The original IEEE 39-bus system had ten generator nodes, twelve connection nodes, seventeen load nodes, and forty-six branches. The parameter settings of the nodes and branches are referred to in [22]. After the improvement, six fresh wind power nodes and six branches were added, and the new wind power node information is shown in Table 1.

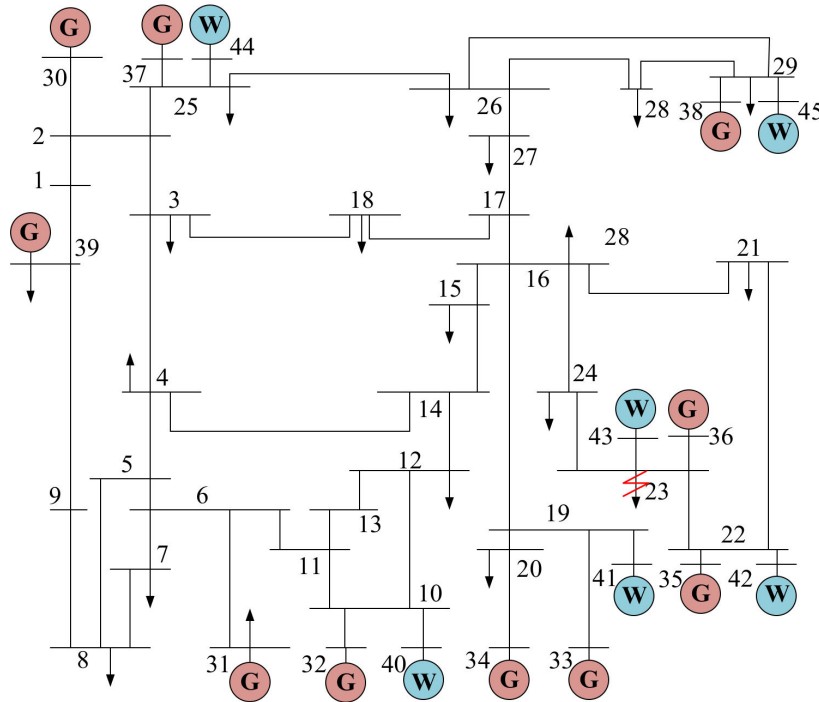

**Figure 9.** The schematic diagram of the improved IEEE 39 node system with wind power.

**Table 1.** The node information of newly added wind power.

| Wind Power Node Number | Branch Number | Initial Ratio of Active Power to Wind Power |
|---|---|---|
| 40 | 40–10 | 3 |
| 41 | 41–19 | 5 |
| 42 | 42–22 | 3 |
| 43 | 43–23 | 2.5 |
| 44 | 44–25 | 2.5 |
| 45 | 45–29 | 4 |

The above simulation system was built using MATLAB PSAT software package (version 2.1.10; Federico Milano, Italy), in which the wind turbine model was a double-fed induction type, and the load was a constant power model. The critical load node 23 was set as an equivalent node. According to the flowchart shown in Figure 8, two analytical value modes of the Thevenin equivalent impedance under different load growth multiples and different wind power penetration $|Z_{eq,1,23}|$, $|Z_{eq,2,23}|$ were calculated, respectively, to study the law of its dynamic change and its relationship with the Black-box Thevenin equivalent impedance mode $|Z_{th,23}|$.

### 4.2. The Dynamic Changes of the Analytical Values of the Equivalent Impedance of Thevenin

Under the premise that the system voltage was stable, the load growth multiple gradually increased from 1.00 to 1.24, and the step size was set to 0.02. The wind power penetration gradually increased from 1% to 50%, and the step size was set to 1%. Then, the three-dimensional diagrams of the dynamic change of the two Thevenin equivalent impedance analytical models of the equivalent node 23 with the load growth multiple and the wind power permeability are shown in Figure 10a,b. Figure 10c is a three-dimensional diagram of the dynamic change of the black-box Thevenin equivalent impedance mode calculated by the two-point method; that is, a small disturbance with a sudden increase or decrease in load power was given to the equivalent node 23. The modulus of the black-box Thevenin equivalent impedance can be obtained by using data of the voltage and current of the node before and after the small power disturbance [10].

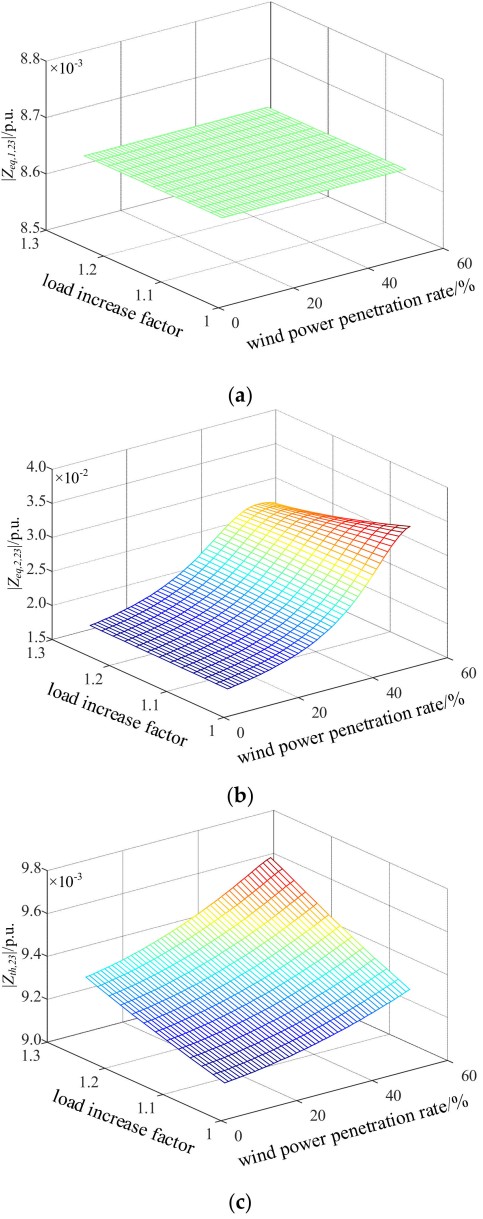

(**a**)

(**b**)

(**c**)

**Figure 10.** The three-dimensional graph of analytical values and accurate value of Thevenin equivalent impedance at the equivalent node 23. (**a**) dynamic changes of $|Z_{eq,1,23}|$; (**b**) dynamic changes of $|Z_{eq,2,23}|$; (**c**) dynamic changes of $|Z_{th,23}|$.

Figure 10a shows that the modulus $|Z_{eq,1,23}|$ of the Thevenin equivalent impedance analytical value of the equivalent mode 1 will always remain unchanged, regardless of the load increase multiple and the wind power penetration rate. $Z_{eq,1,23}$ is the diagonal value of the impedance matrix facing the load node, and the calculation method of the impedance matrix is shown in formula Equation (8). Since the relevant block admittance matrix does not change when the load growth factor and the wind power penetration rate change, $|Z_{eq,1,23}|$ also does not change. Figure 10b shows that $|Z_{eq,2,23}|$ of equivalent mode 2 will decrease with the increase in the load growth multiple, and will first increase and then decrease with the increase in the wind power penetration rate. According to formula Equation (21), when the load growth factor is larger, the self-impedance term plays a leading role in the coupling effect term, that is, the load node 23 is greatly affected by its own injection current and is affected by the coupling effect injection current of the other smaller load nodes, so $|Z_{eq,2,23}|$ becomes smaller. When only the wind power permeability changes, according to the formula in Equation (21), the self-impedance does not change. With the increase in the wind power permeability, the ratio plays a leading role. Therefore, the size of $|Z_{eq,2,23}|$ is determined according to the ratio of the node current of the other load nodes to the node current of the equivalent load node. According to the simulation results, $|Z_{eq,2,23}|$ first increases and then decreases. It can be seen that the ratio of the node current of the other load nodes to the node current of the equivalent load node considering the wind power grid-connected current increment first increases and then decreases. Therefore, the simulation calculation results of the above two Thevenin equivalent impedance analytical values are consistent with the theoretical deduction, which shows the feasibility of the proposed analytical method.

Comparing (a) and (c), (b) and (c) in Figure 10, it can be seen that $|Z_{th,23}|$ is always between $|Z_{eq,1,23}|$ and $|Z_{eq,2,23}|$. $|Z_{eq,1,23}|$ is the lower bound, and $|Z_{eq,2,23}|$ is the upper bound. $|Z_{eq,1,23}|$ plays a dominant role in the Thevenin equivalent impedance. Although the two analytical values of the equivalent impedance of Thevenin do not accurately identify and evaluate the accurate value of the Thevenin equivalent impedance, their upper and lower bounds can be determined, and the impact of wind power access on the Thevenin equivalent impedance can also be theoretically characterized, which has a certain physical significance. When calculating the voltage stability criterion by using the two analytical values of the Thevenin equivalent impedance, not only can the interval range of the Thevenin equivalent impedance mode be determined, but also the influence of the wind power injection current on the voltage stability of the equivalent node can be taken into account. The proposed Thevenin equivalent method can judge the voltage stability of the equivalent node more quickly and accurately and can provide guidance for the judgment of the voltage stability of the new power system.

### 4.3. Comparison of Voltage Stability Criterion and Actual Voltage Phasor

When the voltage is stable, the voltage amplitude and voltage phase angle are within a certain range. Therefore, the actual voltage amplitude and phase angle can be compared with the simulation results of the equivalent method based on specific scenarios to verify the feasibility of the Thevenin equivalent method proposed in this paper. Additionally, when the voltage stability criterion is less than 0, the power system is in an unstable state. Therefore, according to the results, when the voltage stability index calculated by different Thevenin equivalent methods is less than 0, the feasibility of the different methods can be compared and verified. Scenarios 1 and 2 are set up in this section to illustrate that the proposed voltage stability criterion based on the analytical value of the Thevenin equivalent impedance can be more accurate and effective than the black-box Thevenin equivalent model when the load growth factor and the wind power penetration rate change, respectively, to judge the static voltage stability of the equivalent node. Scenario 3 is also set up in this section to illustrate that the proposed criterion can judge the transient voltage stability of the equivalent node more accurately and in a timely manner.

(1)    Scenario 1: load growth multiple changes

When the wind power penetration is 10%, 30% and 50%, the load growth multiple is gradually increased until $L_{23} < 0$, and the voltage stability criteria of the equivalent node 23 are $L_{23}$, $L_{eq,1,23}$ and $L_{eq,2,23}$. The curves of the actual voltage phasor $\dot{U}_{23}$ of node 23 changing with the load growth multiple are shown in Figure 11.

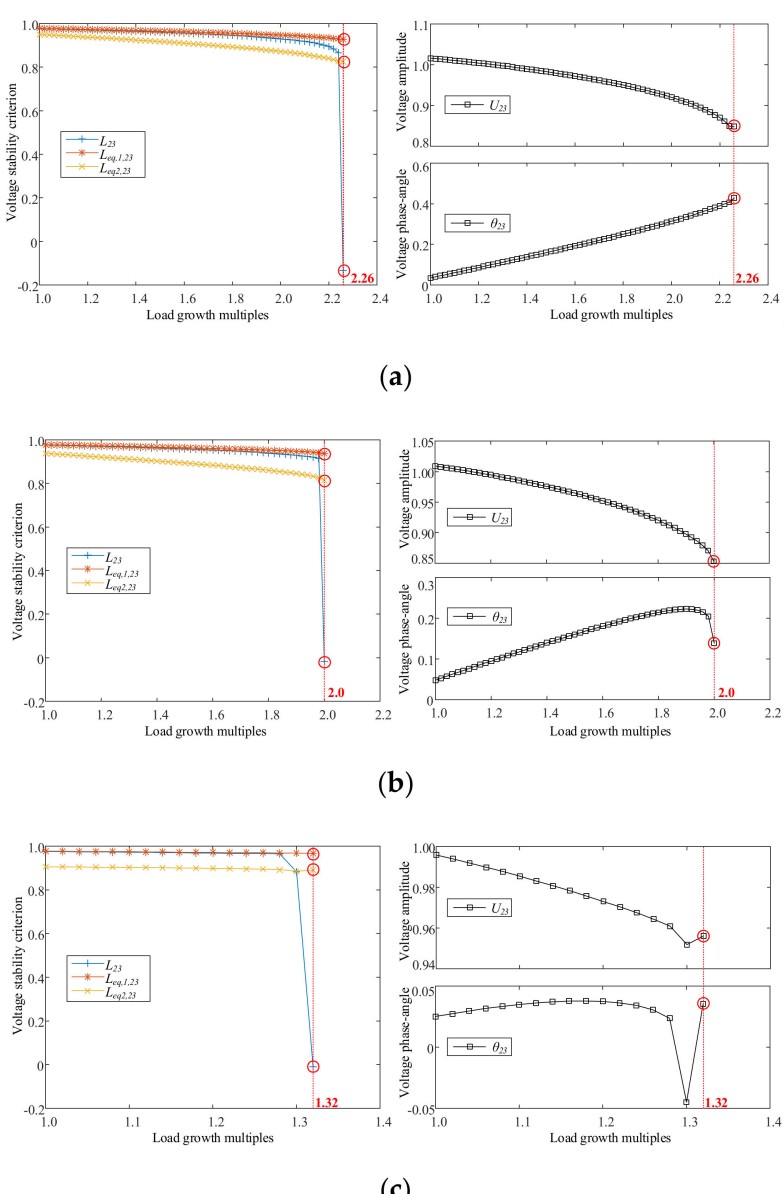

**Figure 11.** The variation curves of voltage stability criterion and actual voltage phasor of the equivalent node 23 when load growth ratio changes. (**a**) System wind power penetration is 10%; (**b**) System wind power penetration is 30%; (**c**) System wind power penetration is 50%.

From Figure 11a–c, it can be seen that the greater the load growth multiple is, the lower the voltage stability of the equivalent is. When the wind power penetration rate of the power system increases from 10% to 50%, the corresponding load growth factor decreases from 2.26 to 1.32 when $L_{23} < 0$; that is, when the set wind power penetration rate is larger, the maximum load that the system can withstand when $L_{23} < 0$ is smaller than the load growth factor. Take Figure 11c as an example, when the load growth factor is 1.32, $L_{23} < 0$. At this time, it should be determined that the static voltage of the equivalent node 23 is unstable when the black-box Thevenin equivalent method is used for calculation. However, at this time, the voltage phasor $\dot{U}_{23}$ of node 23 measured by PMU is $0.9561 \angle 0.0353$, which is still

within a reasonable range, that is, the voltage amplitude $U_{23}$ meets 0.8 p.u. $\leq U_{23} \leq 1.2$ p.u. and the voltage phase angle $\theta_{23}$ meets $-\pi \leq \theta_{23} \leq \pi$. Therefore, for the power system containing wind power, affected by the access of wind power, if the black-box Thevenin equivalent impedance $|Z_{th,23}|$ is used to calculate and judge the voltage stability of the equivalent node, it will be inconsistent with the actual result. However, when the load growth factor is 1.32, the voltage stability criteria $L_{eq,1,23}$ and $L_{eq,2,23}$ calculated by $|Z_{eq,1,23}|$ and $|Z_{eq,2,23}|$ are 0.9669 and 0.8912, respectively, which are greater than 0. It can still be judged that the voltage of node 23 is stable, which is consistent with the actual voltage phasor judgment result at this time. It shows that the two proposed analytical values of the equivalent impedance of Thevenin are effective, which can not only reflect the influence mechanism of the wind power access but can also judge the voltage stability of equivalent nodes more accurately.

(2)    Scenario 2: wind power penetration changes

When the load growth multiples are 1.0, 1.2 and 1.4, the wind power penetration is gradually increased until $L_{23} < 0$, $L_{23}$, $L_{eq,1,23}$ and $L_{eq,2,23}$ and the actual voltage phasor $\dot{U}_{23}$ curves with the wind power penetration are shown in Figure 12.

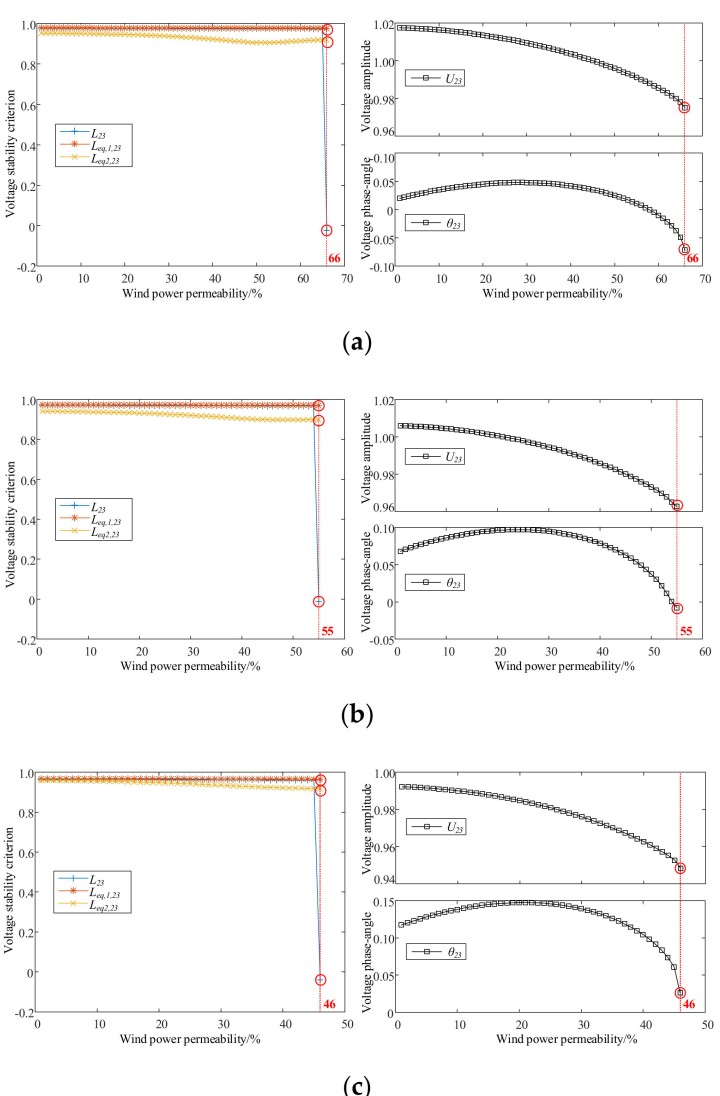

**Figure 12.** The variation curves of voltage stability criterion and actual voltage phasor of the equivalent node 23 when wind power penetration changes. (**a**) The system load growth factor is 1.0; (**b**) the system load growth factor is 1.2; (**c**) the system load growth factor is 1.4.

It can be seen from Figure 12a–c that, if a certain load growth factor is set, the greater the wind power penetration rate, the static voltage stability of the equivalent node 23 will continue to decrease. When the power system load growth factor increases from 1.0 to 1.4, the corresponding wind power penetration rate decreases from 66% to 46% when $L_{23} < 0$. Therefore, the larger the set load increase multiple, the smaller the wind power penetration rate that the system can withstand when $L_{23} < 0$; that is, the larger the set load increase multiple, the smaller the maximum wind power penetration rate that the system can withstand when $L_{23} < 0$. Taking Figure 12c as an example, when the wind power permeability is 46%, $L_{23} < 0$, and the voltage instability of the equivalent node 23 should be determined, but at this time, the voltage phasor $\dot{U}_{23}$ measured by PMU is $0.9482 \angle 0.0261$, which is also within a reasonable range. Therefore, if the black-box Thevenin equivalent impedance $|Z_{th,23}|$ is still used to calculate and judge the voltage stability of the equivalent nodes, it will not be consistent with the actual results and will not apply to the power system with wind power. However, when the wind power permeability is 46%, the voltage stability criteria $L_{eq,1,23}$ and $L_{eq,2,23}$ calculated by $|Z_{eq,1,23}|$ and $|Z_{eq,2,23}|$ are 0.9648 and 0.9193, respectively, which are both greater than 0. It can be judged that the voltage of equivalent node 23 is stable, which is consistent with the actual voltage phasor judgment result. In this case, it also shows the effectiveness of the analytical values of the equivalent impedance of Thevenin obtained by the proposed analytical method, which can more accurately judge the voltage stability of the equivalent node.

(3)　Scenario 3: three-phase short circuit fault

In order to verify the effectiveness of the two Thevenin equivalent methods proposed in this paper under fault conditions, PSAT was used to simulate and set three-phase short-circuit fault conditions, and the Thevenin equivalent calculation results and time domain simulation results were compared and analyzed. When the system load growth multiple is 1.0 and the wind power penetration is 30%, a three-phase short-circuit fault occurs at node 36 at 5 s, and the actual voltage PSAT time-domain simulation curve of the equivalent node 23 is shown in Figure 13.

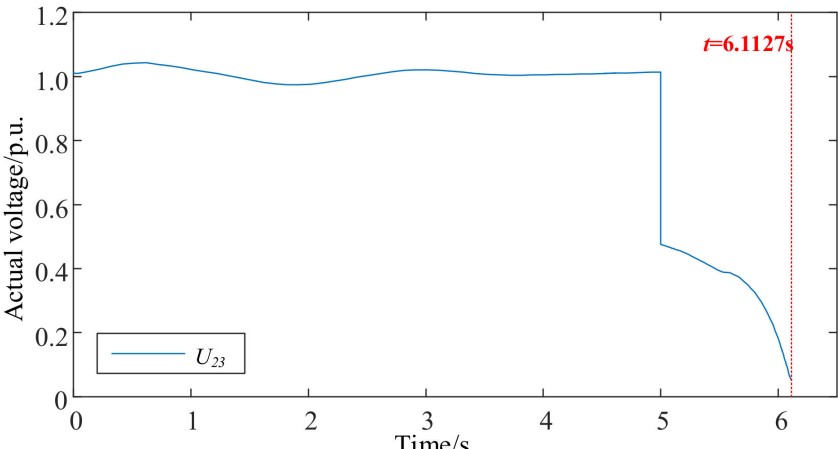

**Figure 13.** The actual voltage time-domain simulation curve of the equivalent node 23.

The PSAT time domain simulation shows that the transient voltage of the equivalent node 23 is unstable at $t = 6.1127$ s. The Thevenin equivalent analytical impedance value is used to calculate the time-varying voltage stability criterion curve of node 23, as shown in Figure 14. Among them, the calculation time of the Thevenin equivalent analytical impedance value is about 0.02472 s, so there is a calculation delay of about 0.02472 s for calculating the voltage stability criterion for each sampling.

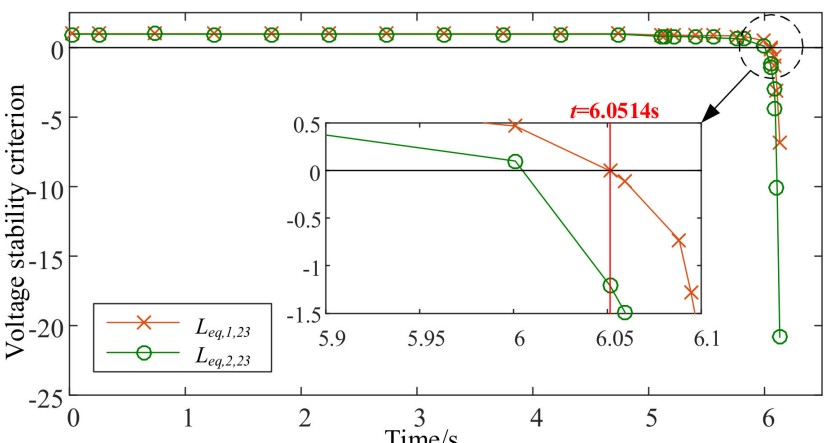

**Figure 14.** The voltage stability criterion curve based on the Thevenin equivalent analytical impedance value of the equivalent node 23.

It can be seen from Figure 14 that when $t$ = 6.0514 s, the voltage stability criteria $L_{eq,1,23}$ and $L_{eq,2,23}$ are less than zero, which can be judged that the transient voltage of equivalent node 23 is unstable. Compared with the PSAT time-domain simulation in Figure 13, it is assumed that the transient voltage of the equivalent node is varying when $t$ = 6.1127 s. The proposed voltage stability criterion based on the Thevenin equivalent analytical impedance can also evaluate the transient voltage stability of the equivalent nodes in time and accurately.

In summary, according to the simulation analysis under different working conditions, it is verified that the two Thevenin equivalent methods proposed in this paper are also applicable to three-phase short-circuit faults, and the method proposed in this paper is applicable to more comprehensive scenarios.

## 5. Conclusions

In this paper, the voltage stability of a power system with wind power was analyzed based on the Thevenin equivalent analytical method. The method considers the current source characteristics of wind power and can analytically express the Thevenin equivalent parameters of the power system with wind power. The voltage stability criterion based on the impedance modulus ratio is calculated by using the analytical value of the Thevenin equivalent impedance. The conclusions are as follows:

(1)  Through the analytical study of the Thevenin equivalent of power systems with wind power, the mechanism of the effect of wind power on the Thevenin equivalent parameters was analyzed. On the one hand, the integration of wind power will change the network physical structure, parameters of the system and the calculation method of the load-oriented impedance matrix, thus affecting the Thevenin equivalent impedance parameters. On the other hand, the injected current of wind power will be converted into the injected current increment of the load node, which will influence the electric potential parameters of the Thevenin equivalent.

(2)  Under different load growth multiples and wind power penetrations, the voltage stability criterion calculated by using the analytical values of two kinds of Thevenin equivalent impedances not only takes into account the influence of wind power access but is also more consistent with the actual voltage phasor results than the voltage stability criterion calculated by using the black-box Thevenin equivalent impedance. In addition, the proposed analytical method can also improve the accuracy and timeliness of the transient voltage stability judgment of equivalent nodes.

However, the analytical calculation of the Thevenin equivalent parameters is not accurate enough in this paper, and the main difficulty lies in the processing of the coupling effect term. Subsequently, the mathematical parameter fitting method or the matrix decoupling

method of the channel component transformation can be combined to obtain accurate analytical results of the Thevenin equivalent parameters. In this way, an accurate voltage stability margin index suitable for a high proportion of new energy power systems can be formulated to study the evaluation method of the maximum penetration rate of new energy in the power grid under the constraint of voltage stability.

**Author Contributions:** Conceptualization, X.Z. and Y.L.; methodology, P.C., Y.L. and T.Z.; software, X.Z. and Y.L.; validation, Y.L., P.C. and F.X.; formal analysis, X.Z.; investigation, Y.L. and X.Z.; resources, X.Z.; data curation, X.Z.; writing—original draft preparation, X.Z. and Y.L.; writing—review and editing, X.Z., F.X. and T.Z.; visualization, X.Z.; supervision, X.Z.; funding acquisition, X.Z., P.C. and F.X. All authors have read and agreed to the published version of the manuscript.

**Funding:** This work was supported by the National Key Research and Development Program of China (2018YFB0904500), the Key Program of the National Natural Science Foundation of China (No. 61933005) and the Natural Science Foundation of Nanjing University of Posts and Telecommunications under Grant (No. NY219094).

**Informed Consent Statement:** Informed consent was obtained from all subjects involved in the study.

**Conflicts of Interest:** The authors declare no conflict of interest.

## Nomenclature

| | |
|---|---|
| PMU | Phasor Measurement Unit |
| TCSC–STATCOM | Thyristor Controlled Series Compensation–Static Synchronous Compensation |
| VRB | Vanadium Redox flow Battery |
| ESS | Energy Storage System |
| $Z_{ik}$ | the impedances between nodes $i, k$ |
| $Z_{kj}$ | the impedances between nodes $k, j$ |
| $\dot{I}_W$ | the current source of wind power connected to the grid |
| $\Delta \dot{I}_i$ | the increment of injected current of the node $i$ |
| $E_{th}$ | the Thevenin equivalent voltage |
| $Z_{th}$ | the Thevenin equivalent impedance |
| $\dot{E}_{th,k}$ | the Thevenin equivalent potential of load node $k$ |
| $Z_{th,k}$ | the Thevenin equivalent impedance of load node $k$ |
| $\dot{U}_{L,k}$ | the voltage phasor of load node $k$ |
| $\dot{I}_{L,k}$ | the current phasor of load node $k$ |
| $Z_{L,k}$ | the load impedance of the equivalent node $k$ |
| $L_k$ | the voltage stability criterion of the equivalent node $k$ |
| $\dot{E}_{Wth}$ | the Thevenin equivalent potential of wind power |
| $Z_{Wth}$ | the Thevenin equivalent impedance of wind power |
| $\dot{E}_{Gth}$ | the Thevenin equivalent potential of synchronous generators of traditional power plants |
| $Z_{Gth}$ | the Thevenin equivalent impedance of synchronous generators of traditional power plants |
| $\boldsymbol{I}$ | the current vectors of all nodes |
| $\boldsymbol{U}$ | the voltage vectors of all nodes |
| $\boldsymbol{Y}$ | the node admittance matrix |
| $G$ | the subscript symbol for generator nodes |
| $W$ | the subscript symbol for wind power nodes |
| $T$ | the subscript symbol for connection nodes |
| $L$ | the subscript symbol for load nodes |
| $\boldsymbol{U}_W$ | the voltage vector of wind power nodes |
| $\boldsymbol{U}_T$ | the voltage vector of connecting nodes |
| $\dot{E}_{open,k}$ | the open-circuit voltage vector oriented to load nodes |
| $\boldsymbol{Z}_{LL}$ | the impedance matrix trained to load nodes |
| $\Delta \boldsymbol{I}_L$ | the injected current increment |

| $Z_{LL,kk}$ | the self-impedance of load node $k$ |
| $\dot{E}_{coupled,k}$ | the coupling effect voltage drop of load node $k$ |
| $\dot{E}_{eq,l,k}$ | the analytical values of the Thevenin equivalent potential of load node $k$ under equivalent mode 1 |
| $Z_{eq,l,k}$ | the analytical values of the Thevenin equivalent impedance of load node $k$ under equivalent mode 1 |
| $\dot{E}_{eq,2,k}$ | the analytical values of the Thevenin equivalent potential of load node $k$ under equivalent mode 2 |
| $Z_{eq,2,k}$ | The analytical values of the Thevenin equivalent impedance of load node $k$ under equivalent mode 2 |
| $Z_{coupled,k}$ | the coupling effect impedance |
| $L_{eq,2,k}$ | the voltage stability criterion of equivalent node $k$ under equivalent mode 2 |

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
