# Peer review of "Voltage Stability Analysis of a Power System with Wind Power Based on the Thevenin Equivalent Analytical Method"

_electronics, doi:10.3390/electronics11111758_

Round 1
Reviewer 1 Report
The Thevenin's theorem states that for any linear electrical network (in the case of interest the networks is in sinusoidal regime) containing only voltage sources, current sources and impedances seen from two chosen terminals can be replaced by an equivalent combination of a voltage source E_eq in a series connection with an impedance Z_eq. The authors seem to refer to a current generator, therefore they might be considering Norton's theorem instead of Thevenin, a confusing exposition of the subject matter.
1. The assumptions under which a wind turbine generator can be considered an ideal voltage source with an impedance connected in series should be better discussed.
2. Check the instability condition on page 3 row 113.
3. In section 2.3 it is said that "the time-varying nonlinearity of the power system in practical applications makes it difficult to identify the Thevenin equivalent parameters accurately". Actually the Thevenin's theorem does not apply to a nonlinear and time varying network; therefore it should be better discussed why the authors consider this an acceptable approximation.
4. The diagram in Fig. 4 shows a current source with two impedances connected to one of the terminals of the same. That is not a Thevenin equivalent, which makes the whole following discussion very confusing.
5. The introduction of the coupling effect voltage drop E_coupled,k and its evaluation is confusing, the proposed discussion and its conclusions are apperently unfruitful.
6. It seems that the results shown in section 4 do not add any novelty to the published literature.
7. In the conclusions, the authors refer to a "Davenin equivalent". Please clarify whether that is a typo.
Reviewer 2 Report
Dear authors, while the presentation is nice in shape, there are a few comments and/or suggestions to improve the manuscript. Please strongly consider the following suggestions:
- Please clarify better the advantages of this paper in the introduction section. I strongly recommend the authors reconsider the related work section for the literature review and discuss the drawbacks of existing literature.
- I suggest that in the Introduction section, the value-added of this paper should be explained. How is it different from other papers in the field? What novelty does it offer? In this sense, in the Reviewed studies on Deployment problem section, the authors must highlight the advantages of the proposed papers with the literature. Please emphasize the real contribution of each paper. Please be more specifically.
- Please also reconsider multiple citation [2-4], [5-7] etc. Please emphasize the contribution of these papers.
- The authors must present the limitation of the proposed approach.
- The Example Analysis section should be revised to improve the impact of the paper. The results are abundant but the analysis is not enough. Please reconsider these aspects and consider in the paper text these values.
- However, I cannot see deep analysis related to them and cannot understand the meaning of the results. Please add more analysis.
- Because the authors use IEEE39 a comparison with other relevant studies is missing.
Minor revisions:
- Please add a list with abbreviations and/or nomenclature, for an easily readable paper.
Round 2
Reviewer 1 Report
The revised version of the paper has answered in an overall satisfactory manner to the comments raised by this reviewer to the first version.
Reviewer 2 Report
The reviewer has no further comments.